# *Mycobacterium bovis* Infection Frequently Requires Surgical Intervention in Individuals with HIV

**DOI:** 10.3390/idr17040082

**Published:** 2025-07-11

**Authors:** Sergio Zuñiga-Quiñonez, Pedro Martinez-Ayala, Monserrat Alvarez-Zavala, Andrea Torres-Rojas, Isaac D. V. Garcia-Govea, Luz A. Gonzalez-Hernandez, Jaime F. Andrade-Villanueva, Fernando Amador-Lara

**Affiliations:** 1Departamento de Clínicas Médicas, Centro Universitario de Ciencias de la Salud, Universidad de Guadalajara, Guadalajara 44280, Mexico; infectologosergio@gmail.com (S.Z.-Q.); pemayala4@gmail.com (P.M.-A.); monse_belan@hotmail.com (M.A.-Z.); luceroga08@gmail.com (L.A.G.-H.); drjandradev@gmail.com (J.F.A.-V.); 2Unidad de VIH, Hospital Civil de Guadalajara Fray Antonio Alcalde, Guadalajara 44280, Mexico; 3Instituto de Investigación en Inmunodeficiencias y VIH, Centro Universitario de Ciencias de la Salud, Universidad de Guadalajara, Hospital 278, Guadalajara 44280, Mexico; andrea.torres2288@alumnos.udg.mx; 4Departamento de Medicina Interna, Instituto Mexicano del Seguro Social, Guadalajara 44329, Mexico; dante.garcia@alumno.udg.mx

**Keywords:** *Mycobacterium bovis*, zoonotic tuberculosis, *Mycobacterium tuberculosis*, HIV infection

## Abstract

Background: Zoonotic infection with *Mycobacterium bovis* continues to occur, particularly in regions lacking bovine tuberculosis surveillance and where the consumption of unpasteurized dairy products, including artisanal cheeses, is common. We describe the clinical and microbiological characteristics, diagnostic procedures, and treatment outcomes of individuals with HIV with *M. bovis* infection. Methods: We conducted a retrospective study analyzing sociodemographic, clinical, microbiological, and computed tomography (CT) data, as well as treatment outcomes, in 12 patients with HIV with confirmed *M. bovis* infection. These findings were compared with those of 14 individuals with HIV diagnosed with *Mycobacterium tuberculosis* infection during the same period. Results: Consumption of unpasteurized dairy products was significantly associated with *M. bovis*. Patients with *M. bovis* infection had higher CD4+ T-cell counts compared to those with *M. tuberculosis* infection (*p* = 0.01, *r* = 0.45). All *M. bovis* cases presented with extrapulmonary disease. CT imaging in *M. bovis* infection more frequently demonstrated retroperitoneal lymphadenopathy, hepatosplenomegaly, and splenic abscesses compared to *M. tuberculosis* infection. Microbiological identification was exclusively from extrapulmonary sites in all *M. bovis* cases. Surgical interventions, including abscess drainage or splenectomy, were significantly more common among *M. bovis* patients. Conclusions: *M. bovis* infection in individuals with HIV is characterized by consistent extrapulmonary, often abdominal, involvement. Surgical procedures are frequently required for both diagnosis and management. Targeted efforts to identify *M. bovis* are warranted, particularly in high-burden regions where unpasteurized dairy consumption remains prevalent.

## 1. Introduction

*Mycobacterium bovis*, a member of the *Mycobacterium tuberculosis* complex (MTBC), is the principal cause of zoonotic tuberculosis (zTB), a form of human tuberculosis [1]. Infection with *M. bovis* in humans remains a neglected public health issue, particularly in regions where bovine tuberculosis is not effectively controlled [2]. While estimates of zTB incidence and mortality remain imprecise, it is believed that approximately 140,000 new cases and 11,400 deaths occur annually worldwide [3]. Globally, zTB accounts for around 1.4% of all tuberculosis cases, though this figure is likely underestimated, especially in resource-limited settings where systematic surveillance is lacking and diagnostic capabilities to differentiate between *M. tuberculosis* and *M. bovis* are limited [4]. Accurate identification requires mycobacterial culture followed by biochemical or molecular diagnostics, which are often unavailable in these settings [5].

The consumption of unpasteurized milk and dairy products remains the primary route of *M. bovis* transmission to humans [6]. In many low- and middle-income countries, milk pasteurization is inconsistently implemented. In Mexico, for instance, nearly 30% of milk is sold unpasteurized, including that used for artisanal cheese production [7]. The artisanal cheese-making process, which often relies on raw milk, does not eliminate viable *M. bovis*, and several studies have detected the bacillus in these products using culture and molecular techniques [8].

Bovine tuberculosis (bTB) is endemic in several parts of the world. Data from 119 countries indicate that *M. bovis* circulates in 59% of national cattle herds, yet only 10% of these countries have implemented zTB-specific surveillance [9]. The absence of national bTB control programs, particularly those involving test-and-slaughter policies, further increases the risk of zoonotic transmission to humans [10].

Although human-to-human transmission of *M. bovis* is believed to be less efficient than that of *M. tuberculosis*, due in part to mutations in the PhoP/PhoR two-component virulence regulatory system [11], outbreaks of *M. bovis* have been reported, including multidrug-resistant strains, particularly among individuals with HIV, in whom high mortality rates have been observed [12,13].

In many low-resource settings, tuberculosis diagnosis still relies on sputum smear microscopy or rapid molecular assays, such as Xpert MTB/RIF^®^, which is widely endorsed by the World Health Organization as a major advancement in TB diagnostics. Although Xpert MTB/RIF^®^ is effective for detecting both pulmonary and extrapulmonary TB, as well as rifampicin resistance via rpoB gene mutations [14], it cannot distinguish between *M. tuberculosis* and *M. bovis*. Consequently, diagnostic algorithms based solely on Xpert MTB/RIF^®^ positivity without subsequent species identification by culture or molecular testing may lead to underdiagnosis and misclassification of *M. bovis* infections [15]

This study aims to describe the clinical and microbiological features, computed tomography (CT) imaging findings, surgical interventions, and treatment outcomes of individuals with HIV with *M. bovis* infection, in comparison with those infected with *M. tuberculosis* during the same period.

## 2. Materials and Methods

### 2.1. Design

This retrospective study was conducted in Guadalajara, Jalisco, Mexico, between January 2019 and March 2023. Data was obtained from the database of the HIV Unit at the Hospital Civil de Guadalajara, with approval from the hospital’s Ethics Committee (protocol code: CEI 132/25, date of approval: 13 January 2025).

We collected sociodemographic data, comorbidities, clinical and microbiological characteristics (including symptoms, anatomical site of infection, site of isolation/identification, and diagnostic method), computed tomography (CT) findings, surgical interventions, and treatment outcomes from individuals with HIV diagnosed with *M. bovis* infection confirmed by culture and/or polymerase chain reaction (PCR). These data were compared with those from patients with HIV diagnosed with *M. tuberculosis* infection during the same period, also confirmed by culture and PCR.

Nucleic acid amplification testing using the Xpert^®^ MTB/RIF assay (manufactured by Cepheid, Sunnyvale, CA, USA) was employed for the diagnosis of *M. tuberculosis* complex infection. For the detection of *M. bovis* infection, multiplex polymerase chain reaction (PCR) was performed using the MYCODirect 1.7 LCD-Array kit^®^ (manufactured by Chipron Technologies GmbH, Berlin, Germany).

Patients with *M. tuberculosis* infection received treatment in accordance with the World Health Organization’s (WHO) tuberculosis treatment guidelines [16]. For those diagnosed with *M. bovis* infection, a fluoroquinolone (levofloxacin) was included in the treatment regimen due to the intrinsic resistance of *M. bovis* to pyrazinamide.

### 2.2. Statistical Analysis

The sample size was determined through a retrospective review of confirmed *M. bovis* infection cases diagnosed between 2019 and 2023 at a specialized HIV care center serving patients from multiple western states of Mexico. *M. bovis* remains an under recognized and neglected pathogen in many regions, largely due to limitations in diagnostic infrastructure. For comparative analysis, an equivalent number of confirmed *M. tuberculosis* cases were included. Consequently, the final sample size was defined by the total number of *M. bovis* cases identified during the study period.

Categorical variables were summarized using frequencies and percentages, while continuous variables were described using either medians and interquartile ranges (IQR) or means and standard deviations (SD), as appropriate. Qualitative variables were analyzed using Fisher’s exact test. For quantitative variables, Student’s *t*-test was used for parametric data, and the Mann–Whitney *U* test was applied for non-parametric data. To assess the effect size for categorical data, the Odds Ratio (OR) was calculated to estimate the strength of association between exposure and outcome. In the case of non-parametric ordinal data, Rosenthal’s *r* was used as a measure of effect size. All statistical analyses were performed using RStudio, version 4.5.2 (Posit PBC, Boston, MA USA) and IBM SPSS Statisitics version 29 (IBM Corp, Armonk, NY, USA), GradPack30 (IBM Corp., Armonk, NY, USA, Sku: 44V5829-1-1-1). Details of the prompts used are available in the Appendix A. A *p*-value < 0.05 was considered statistically significant.

## 3. Results

Twelve cases of *M. bovis* infection were identified during the study period. These cases were initially classified as *M. tuberculosis* infections due to positive results from the Xpert^®^ MTB/RIF assay performed on samples from various anatomical sites. However, all were subsequently confirmed as *M. bovis* infections through multiplex PCR or culture. Clinical and demographic characteristics of these cases were compared with fourteen cases of *M. tuberculosis* infection, diagnosed by culture during the same period at the same hospital.

Two significant differences were observed between the groups: a higher frequency of habitual consumption of unpasteurized dairy products—recognized as a risk factor for *M. bovis* infection—and a significantly higher median CD4+ T-cell count in patients with *M. bovis* infection compared to those with *M. tuberculosis* (*p* < 0.0001 and *p* < 0.01, respectively) (Table 1). Although unpasteurized dairy consumption showed a statistically significant association, we attempted to assess its size effect; however, this was not feasible, as there were no reported cases in the *M. tuberculosis* group. The lack of variability in this variable prevented a proper interpretation of the effect size. For the CD4+ T-cell variable, an effect size of r = 0.45 was obtained using Rosenthal’s r formula, indicating a moderate to large effect and suggesting that the difference between groups is not only statistically significant but also clinically meaningful. Notably, at the time of diagnosis, 10 out of 12 patients (83.3%) with *M. bovis* infection were not receiving antiretroviral therapy (ART) three were newly diagnosed and ART-naïve, while seven had discontinued treatment. Similarly, 11 out of 14 patients (85.7%) with *M. tuberculosis* infection were not on ART, including six newly diagnosed individuals and five with a history of treatment abandonment. This highlights the vulnerability of untreated or irregularly treated HIV-positive individuals to both forms of mycobacterial infection.

No significant differences were observed in clinical symptoms between the two groups, except for neurological symptoms, which were significantly more frequent in *M. tuberculosis* infections compared to *M. bovis* infections (*p* < 0.03). Pulmonary involvement predominated in *M. tuberculosis* infection (*p* < 0.01). Notably, all *M. bovis* cases (100%) exhibited extrapulmonary disease. Several abdominal computed tomography (CT) findings were significantly more frequent in patients with *M. bovis* infection compared to those with *M. tuberculosis*, including retroperitoneal lymphadenopathy (*p* = 0.004), hepatomegaly (*p* = 0.001), splenomegaly (*p* = 0.0003), and splenic abscesses (*p* = 0.004).

The anatomical sites of microbiological isolation or molecular identification also differed between groups. *M. tuberculosis* was more frequently isolated from pulmonary sites, whereas *M. bovis* showed no pulmonary isolation (*p* = 0.001) (Table 2). An odds ratio (OR) analysis was performed to evaluate the association between selected clinical variables and the presence of *M. bovis* infection. This statistical method allowed for the estimation of the strength of association between potential risk factors and the outcome of interest. Variables showing significant associations with *M. bovis* included retroperitoneal lymphadenopathy (OR = 15.7, 95% CI: 1.94–227.5) and combined pulmonary and extrapulmonary involvement (OR = 15.4, 95% CI: 1.9–224.3). These results highlight the relevance of these clinical features in the differential diagnosis of *M. bovis* infection.

Positive cultures were obtained in only 4 of the 12 cases (33.3%) of *M. bovis* infection; the remaining cases were identified by molecular methods. In contrast, all 14 cases of *M. tuberculosis* infection were confirmed by positive cultures. Due to the predominant extrapulmonary involvement in *M. bovis* cases, surgical intervention was required in 8 patients (66.6%) for diagnostic and therapeutic purposes, including percutaneous catheter placement (*n* = 2), open surgery (*n* = 6), and splenectomy (*n* = 5), primarily for drainage or excision of abdominal abscesses. No surgical procedures were performed in patients with *M. tuberculosis* infection (*p* < 0.0003). There were no statistically significant differences between the two groups in terms of treatment failure or mortality (Table 3).

## 4. Discussion

Our retrospective comparative study of patients with HIV demonstrates marked differences in clinical manifestations between *M. bovis* and *M. tuberculosis* infection. The pronounced abdominal tropism of *M. bovis* aligns with its typical alimentary transmission, the ingestion of contaminated animal products leads to gastrointestinal and disseminated TB rather than the primary pulmonary disease seen with inhaled *M. tuberculosis* [17,18,19]. Prior studies in endemic regions have likewise found *M. bovis* to be significantly associated with extrapulmonary TB (adjusted OR~1.8 for extrapulmonary involvement) [17] and with abdominal lymph node and organ infection in immunocompromized hosts [18]. Consistent with these findings, the computed tomography (CT) profiles of our *M. bovis* patients frequently showed abdominal abscesses and lymphadenopathy. Consequently, invasive surgical interventions (such as image-guided abscess drainage, exploratory laparotomy, or splenectomy) were often required for diagnosis and treatment in *M. bovis* cases, whereas none of the *M. tuberculosis* cases needed such procedures. These findings highlight the severe and localized nature of *M. bovis* infection within the abdomen and underscore the diagnostic challenge it poses when easily accessible specimens, such as sputum, are unavailable. Moreover, the frequent observation of retroperitoneal lymphadenopathy, hepatosplenomegaly, and splenic abscesses on CT scans in *M. bovis* cases should prompt clinicians to consider a zoonotic etiology in the differential diagnosis [19].

Notably, immunological status differed between the two groups. Patients with *M. bovis* infection had significantly higher CD4+ T-cell counts on presentation compared to those with *M. tuberculosis*. This finding suggests that zTB can occur in individuals with HIV before profound immunosuppression develops. It contrasts with earlier observations that linked *M. bovis* disease to advanced AIDS—for example, a study in California reported that HIV patients with *M. bovis* TB were usually severely immunosuppressed (CD4 ≤ 200 cells/μL) [18] and frequently had abdominal involvement. Our data implies that *M. bovis* infection may occur at higher CD4+ levels than typically observed in *M. tuberculosis* reactivation. This seemingly paradoxical observation could reflect differences in the timing of HIV diagnosis, antiretroviral therapy initiation, access to highly active antiretroviral therapy (HAART), or delays in species-level identification due to limited molecular diagnostic capabilities [4]. Importantly, this observation raises critical questions regarding the immunopathogenesis of *M. bovis* in immunocompromized hosts and warrants further investigation. Importantly, a striking proportion of patients in both groups were not receiving ART at the time of diagnosis—83.3% in the *M. bovis* group and 85.7% in the *M. tuberculosis* group—including newly diagnosed individuals and those with treatment abandonment. This pattern highlights the continued vulnerability of ART-naïve or non-adherent individuals to opportunistic infections, even at moderately preserved CD4 counts. Recent studies suggest that immune dysregulation in ART-naïve individuals extends beyond CD4+ depletion, involving sustained inflammation, monocyte activation, and impaired pathogen containment [13]. Moreover, delayed ART initiation or treatment interruptions have been associated with increased risk for extrapulmonary and disseminated TB [20]. Thus, the absence of ART may predispose to *M. bovis* infection not necessarily through severe immunosuppression, but via qualitative immune dysfunction and unrestrained bacterial dissemination. Consistent with our findings, a large cohort study in Mexico City reported a higher prevalence of *M. bovis* among non-HIV immunosuppressed individuals compared to those living with HIV (54% vs. 36% of extrapulmonary/disseminated TB cases, respectively) [20]. These findings underscore that severe immunodeficiency is not a strict requirement for *M. bovis* disease; moderate immunosuppression or other risk factors may be sufficient in the context of exposure. Nevertheless, HIV remains a significant risk factor for acquiring zTB overall [20].

Our findings also reinforce the critical role of zoonotic exposure in driving *M. bovis* infections. A history of unpasteurized dairy consumption was strongly associated with *M. bovis* TB in our patients, consistent with the classical route of transmission from cattle. In regions like Mexico, about a quarter of the milk is sold unpasteurized [7], and artisanal cheeses made from raw milk have repeatedly been shown to harbor viable *M. bovis* bacilli [8]. A whole-genome sequencing study in Baja California provided direct evidence of this food-borne pathway: the genotypes of *M. bovis* isolated from local cattle, unpasteurized cheese, and human TB patients were nearly identical, indicating ongoing cross-species transmission in the dairy supply chain [8]. This exemplifies a clear One Health scenario where bovine tuberculosis in livestock propagates into human disease [9]. Similarly, epidemiological data from the United States show that a disproportionate number of *M. bovis* cases occur among people of Hispanic origin and have been traced to cheese made from unpasteurized milk imported from Mexico [8]. In our cohort, 75% of those with *M. bovis* infection reported regular consumption of raw milk or artisanal cheese, bolstering the link between dietary exposure and disease. Other known risk factors for zTB include close contact with cattle, younger age, and immunosuppressive therapies [17,21]. Taken together, these patterns underscore that preventable exposures—especially ingestion of contaminated dairy—remain a key driver of *M. bovis* infection in humans.

A major challenge illuminated by our study is the diagnostic gap in recognizing *M. bovis*. Clinically and radiographically, tuberculosis caused by *M. bovis* is virtually indistinguishable from that caused by *M. tuberculosis* [17]. All patients in our series were initially diagnosed with “tuberculosis” via positive Xpert MTB/RIF assays, which detect the *M. tuberculosis* complex but cannot differentiate species [14]. It was only through subsequent culture or PCR-based speciation that *M. bovis* was confirmed in the 12 cases. This scenario is likely common in many settings: if species identification is not routinely performed, zTB cases become misclassified as ordinary TB. Indeed, a 2021 meta-analysis found the reported proportion of *M. bovis* among human TB cases to vary enormously—from as low as 0.4% in some studies to as high as 76% in others—depending on the diagnostic methods employed [22]. Such variability highlights how under-detection of *M. bovis* is a widespread problem when surveillance and lab capacity are limited [4]. In many low-resource regions, TB diagnosis still relies on sputum smear or Xpert, and even when there are cultures, the isolates may not be speciated [5]. Accurate identification of *M. bovis* requires mycobacterial culture followed by biochemical or molecular tests, which are often unavailable outside reference laboratories [23,24]. There are also technical impediments: the standard Löwenstein–Jensen culture medium (with glycerol) inhibits *M. bovis* growth, yet many labs rely on it by default [22]. A glycerol-free pyruvate-containing medium (e.g., Stonebrink) is recommended for isolating *M. bovis*, but it is seldom used in practice [25]. Newer liquid culture systems (e.g., MGIT 960) improve overall mycobacterial detection, but even when *M. bovis* grows, additional time-consuming tests are needed for species confirmation [26]. Various PCR-based assays (including multiplex and lineage-specific tests) have been developed to differentiate *M. bovis*, for instance by targeting the pncA gene or regions of difference, but these advanced tools remain largely inaccessible in the resource-limited settings where zTB is most prevalent [23]. Collectively, these diagnostic challenges contribute to a persistent underestimation of *M. bovis* in human TB. They also hinder timely, appropriate therapy for affected patients.

The need for surgical intervention in 66.7% of *M. bovis* infections observed in our study underscores a significant diagnostic and logistical challenge. This high rate is likely attributable to the frequent extrapulmonary localization of lesions, which are often inaccessible through non-invasive methods. Although this percentage appears striking, it should be interpreted with caution due to the limited number of subjects included in the study. Previous reports have documented *M. bovis* isolation from pulmonary, extrapulmonary, or both compartments, with the abdomen and central nervous system frequently serving as primary sites—areas that typically require invasive procedures for definitive microbiological identification [17].

From a public health perspective, our study underlines the need for greater attention to zTB in both the animal and human health sectors. Most low- and middle-income countries lack specific surveillance programs for *M. bovis* in humans [27], and many have only partial control programs for bovine tuberculosis in cattle [28]. As of a recent survey, *M. bovis* is present in cattle herds in most countries, yet few countries have implemented dedicated zTB surveillance and reporting systems [9]. This gap in surveillance means that the true burden of *M. bovis* in communities is poorly quantified and likely far higher than official figures suggest [2]. The absence of robust bovine TB control (such as test-and-slaughter eradication programs with compensation for farmers) in endemic areas further perpetuates spillover transmission to humans [10]. Our findings support calls for a comprehensive One Health approach [9] to break this cycle. This approach would include strengthening veterinary public health measures (controlling bovine TB in livestock, enforcing milk pasteurization, and monitoring animal reservoirs) alongside improvements in clinical care (better diagnostics and treatment for zTB). These challenges highlight the urgent need for improved surveillance, accessible diagnostic infrastructure, and heightened clinical awareness to address the true burden of zTB [4]. Differentiating *M. bovis* from *M. tuberculosis* in patients—especially those with HIV or other immunosuppression—is not only epidemiologically important but also clinically crucial. *M. bovis* is intrinsically resistant to pyrazinamide (PZA), a first-line TB drug [17]. Standard 6-month short-course therapy for TB relies on PZA in the initial two months to shorten and sterilize treatment; if an *M. bovis* infection is mistakenly treated as ordinary TB, the ineffective PZA could lead to suboptimal outcomes [25]. In our cohort, recognizing the species allowed us to modify therapy (adding a fluoroquinolone in place of PZA) as per current guidelines, and this likely contributed to good outcomes. While mortality in disseminated *M. bovis* can be high in advanced HIV disease [20], only one patient with *M. bovis* (8.3%) in our series died, possibly reflecting the benefit of timely surgical intervention combined with appropriate drug therapy. This favorable result, albeit in a small sample, underscores that with proper management, *M. bovis* infection in HIV-positive individuals is treatable. The overarching challenge remains to make such targeted diagnosis and treatment accessible wherever co-infection of HIV and zTB arises. Limitations of our study include its retrospective design, small sample size, and exclusive inclusion of patients with HIV, which may limit generalizability. In particular, the high proportion of surgical interventions observed should be interpreted with caution, as it may have been influenced by the limited number of cases analyzed.

## 5. Conclusions

*M. bovis* infection remains underdiagnosed due to limited surveillance of zTB and restricted laboratory capacity, posing a significant public health challenge. Individuals with HIV are particularly susceptible to mycobacterial infections, making the identification of *M. bovis* especially relevant. Extrapulmonary involvement is a key feature in diagnosing *M. bovis* infection among individuals with HIV, and surgical interventions are often required for both diagnostic and therapeutic purposes. Improving diagnostic capabilities and ensuring timely, appropriate treatment are essential, particularly in low-resource settings with a high burden of *M. bovis* infection.

## Figures and Tables

**Table 1 idr-17-00082-t001:** Sociodemographic characteristics of patients with HIV with *Mycobacterium* bovis vs. Mycobacterium *tuberculosis* infection.

Characteristics		*M. bovis* (*n* = 12)	*M. tuberculosis* (*n* = 14)	*p* Value
Mean age, years (SD)		39.64 ± 8.86	38.3 ± 9.25	ns
Gender	Male	11 (92%)	14 (100%)	ns
	Female	1 (8%)	0 (0%)	
Regular consumption of unpasteurized dairy products (milk, artisan cheeses)		9 (75%)	0 (0%)	<0.0001
Recent contact with individuals with TB		1 (8.3%)	2 (14.3%)	ns
Alcohol consumption		6 (50%)	7 (50%)	ns
Current smoking		4 (33.3%)	10 (71.4%)	ns
Diabetes mellitus		1 (8.3%)	0 (0%)	ns
Charlson Comorbidity Index		6.25	6.2	ns
CD4+ T-cell count (cells/μL), median (IQR)		102.5 (42–155)	26 (15.75–47)	0.02
HIV-1 RNA (copies/mL), median (IQR)		176,260 (277–266,250)	175,009.5 (106,826.2–656,014.2)	ns

Abbreviations: SD, standard deviation; IQR, interquartile range; ns, not significant. Qualitative variables were analyzed using Fisher’s exact test. Quantitative variables were analyzed using Student’s *t*-test for parametric data (e.g., age) and the Mann–Whitney *U* test for non-parametric data (e.g., CD4+ T-cell count and HIV-1 RNA levels). A *p*-value < 0.05 was considered statistically significant.

**Table 2 idr-17-00082-t002:** Clinical, imaging, and microbiological characteristics of patients with HIV with *Mycobacterium* bovis vs. Mycobacterium *tuberculosis* infection.

Characteristics	*M. bovis* (*n* = 12)	*M. tuberculosis* (*n* = 14)	*p* Value
Presenting symptoms			
Fever	10 (83.3%)	8 (57.1%)	ns
Cough	8 (66.7%)	6 (42.3%)	ns
Weight loss (>10%)	7 (58.3%)	11 (78.6%)	ns
Cervical lymphadenopathy	11(91.7%)	10 (71.4%)	ns
Gastrointestinal (abdominal pain, diarrhea, vomiting)	8 (66.7%)	12 (85.7%)	ns
Neurological	1 (8.3%)	7 (50%)	0.03
Anatomical sites of involvement			
Pulmonary (only)	0 (0%)	6 (42.9%)	0.01
Extrapulmonary (only)	5 (41.7%)	3 (21.4%)	ns
Pulmonary and extrapulmonary	7 (58.3%)	6 (42.9%)	ns
Pulmonary CT findings			
Miliary	4 (33.3%)	5 (35.7%)	ns
Cavitations	1 (8.3%)	3 (21.4%)	ns
Bronchiectasis	1 (8.3%)	1 (7.1%)	ns
Pleural	2 (16.7%)	0 (0%)	ns
Abdominal CT findings			
Retroperitoneal lymphadenopathy	10 (83.3%)	3 (21.4%)	0.004
Psoas abscess	4 (33.3%)	2 (14.3%)	ns
Hepatomegaly	7 (58.3%)	0 (0%)	0.001
Splenomegaly	8 (66.7%)	0 (0%)	0.0003
Splenic abscesses	6 (50%)	0 (0%)	0.004
Site of isolation/molecular identification			
Pulmonary †	0 (0%)	7 (50%)	0.001
Neck lymph nodes	6 (41.7%)	4 (25%)	ns
Abdominal ‡	5 (33.3%)	4 (28.6%)	ns
Genitourinary	0 (0%)	0 (0%)	ns
Bones, joints, skin, and soft tissues	2 (16.7%)	1 (7.1%)	ns

Abbreviations: CT, computed tomography; ns, not significant. Qualitative variables were analyzed using Fisher’s exact test. A *p* value < 0.05 was considered significant. † Includes sputum, bronchoalveolar lavage, gastric aspirate samples. ‡ Includes peritoneal/retroperitoneal fluid, liver, spleen, psoas abscess and stool samples.

**Table 3 idr-17-00082-t003:** Surgical procedures and outcomes in patients with HIV with *Mycobacterium* bovis vs. Mycobacterium *tuberculosis* infection.

Characteristics	*M. bovis* (*n* = 12)	*M. tuberculosis* (*n* = 14)	*p* Value
Surgical procedure †	8(66.7%)	0 (0%)	<0.001
Outcomes			
Cured	8 (66.6%)	7 (50%)	ns
Treatment failure	2 (16.7%)	2 (14.3%)	ns
Lost to follow-up	1(8.3%)	3 (21.4%)	ns
Death	1 (8.3%)	2 (14.3%)	ns

Qualitative variables were analyzed using Fisher’s exact test. Quantitative variables were analyzed using Mann–Whitney *U* test. A *p*-value < 0.05 was considered statistically significant. † Surgical procedures were performed for diagnostic and therapeutic purposes, including percutaneous catheter placement, open abdominal surgery, or splenectomy for drainage/removal of abscesses.

## Data Availability

All relevant data are within the paper.

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
