# Peer review of "Mycobacterium bovis Infection Frequently Requires Surgical Intervention in Individuals with HIV"

_2036-7449, 2025, doi:10.3390/idr17040082_

Round 1

Reviewer 1 Report

Comments and Suggestions for Authors

The short quite interesting manuscript addressed to Infectious Disease Report by dr. Fernando Amador-Lara and colleagues entitled “M. bovis infection: high requirement for surgical interventions in HIV-infected subjects” is like to no actual because authors cite one original work [ref. (32)] from last three years [2023–2025] and only 5 original works from years 2020–2022.

The authors analyzed in detail the clinical, microbiological, treatment, and outcome characteristics of patients infected with M. bovis compared with those infected with M. tuberculosis, and the analysis included patients infected with HIV. As the authors report in the Conclusions section, M. bovis infection is underdiagnosed due to the lack of routine veterinary epidemiological surveillance and limited laboratory capacity. Proper identification and timely and appropriate treatment pose a major public health challenge, especially in resource-limited regions with a high burden of M. bovis infection, especially in HIV-infected humans who are more susceptible to mycobacterial infections.

The manuscript should be published at Infectious Disease Reports Journal because of its possible impact on government funding decisions for such pressing problems, but only after certain shortcomings listed below have been addressed:

·       The cited literature will raise suspicions about the obsolescence of the problem that is the subject of the manuscript, which may be an effective reason for budget cuts, not for the expected increase in subsidies. Therefore, the manuscript should definitely be updated with contemporary literature from the research topic and/or from related fields according to the authors' discretion. Please cite and critically discuss original work from the years (2020–2025), preferably from the last three years (2023–2025). ·       At line 51 is … cheese(8) … , but should be … cheese (8) … . Comment: Please add a space before the bracket.

  • At line 52 is … 80-90 … , but should be better … 80–90 … . Comment: Nowadays in scientific article the medium sign " – " is preferred between numbers. See line such 56.

  • At line 53 is … communities(9) … , but should be … communities (9) … . Comment: Please add space before the bracket.

  • At line 91 is … guidelines(22) … , but should be … guidelines (22) … . Comment: Please add space before the bracket.

  • At lines 114–115 (at body of table 1) and at line 238 is … 38-165 … , … 14.5-43 … , … 229-300750 … , … 74600-683264 … , … 2.6-8.3 … , but should be better … 38–165 … , … 14.5–43 … , … 229–300750 … , … 74600–683264 … , … 2.6–8.3 … , respectively. Comment: Nowadays in scientific article the medium sign " – " is preferred between numbers. See line such 296.

·       At line 120, 122–123 and 126 are … (p <0.03) … , … (p <0.01) … , … (p<0.001) … , respectively, but please unify the style in the manuscript content.

·       At line 125 is … (p=0 0.01), … , but please check the numerical value and correct accordingly.

  • At lines 126 and 130 are … (p=0.001) … , … (p =0.01) … , respectively, but please standardize the style at the body of the manuscript.

  • At line 166 is … globally(5). … , but should be … globally (5). … .

  • At line 179 is … (25) … , but should be … (25). … . Comment: Please add the end dot mark.

  • At line 183 is … gene(26) … , but should be … gene (26) … . Comment: Please add the space.

  • At line 242 is … (39) Contrary … , but should be … (39). Contrary … . Comment: Pleas add the dot mark.

  • At line 247 is … 0.006)(42) … , but should be … 0.006) (42) … . Comment: Pleas add the space mark. See line such 234.

  • At line 293 is … Global tuberculosis report 2019. ]. … , but please check style and correct. Comment: Please synchronize the style at the body of the manuscript.

Reviewer 2 Report

Comments and Suggestions for Authors

The authors present a retrospective study a HIV-infected cohort of patients diagnosed with either Mycobacterium bovis or M. tuberculosis infections as co-morbidities. The study identifies the consumption of raw milk and its derivatives and high CD4+ T cell counts as risk factors M. bovis infection compared to M. tuberculosis infection. Similarly those diagnosed with M. bovis infection were more likely to require surgical intervention, compared to those with M. tuberculosis infections. While the study involves a limited number of patients, this is acknowledged and authors do not over interpret their results.

The introductory text is relevant and provides the required information to understand the subsequent study. The literature cited is relevant and useful. I have asked the authors to provide additional references for some statements of fact.

The methods are described in sufficient detail to enable replication of the study.

The results are clear presented, with the descriptions in the main text matching what is shown in the associated tables. I would suggest that the authors ensure their descriptions including the biological impacts and not just

The discussion is long and contains some paragraphs that summarise the reported findings of published studies, but do explain why the study and its data are relevant in the context of the data presented in the current study. I would suggest that the authors review the discussion and ensure that, where the details of other studies are described, it is done so in the context of the findings of their study and its data. If these links cannot be articulated, then deletion of the paragraph should be considered.

Comments and suggestions

Line 2 suggest revision of “M. bovis” to “Mycobacterium bovis”

It is not ideal to have abbreviations in the title. Though I think “HIV” is acceptable given its commonality of use.

I would also suggest that the authors consider revising the title, “high requirement”

“Mycobacterium bovis infection increases the risk of surgical interventions in HIV-infected patients.”

Or something along these lines. Though the authors need to determine the most important finding of their study, is the risk factors associated with M. bovis infection or the increased risk of the need for surgical interventions? I think it is probably the need the for “surgical interventions”, so the authors may although considered this.

Noting that “requirement” is spelled incorrectly in the current title.

Comments and suggestions.

Line 18 suggest revision “of HIV-infected subjects with M. bovis infection.”

Line 21 suggest revision "which compared”

Line 25 I would suggest the authors add the details of these factors. While p values are somewhat useful, they do not provide any details on the magnitude of the effect, this statistical robustness without the biological context are of limited value.

Line 26 suggest revision “CT findings in M. bovis infected patients were deemed to be significant more frequently than those with M. tuberculosis infection, including the detection of retroperitoneal lymphadenopathy,”

Line 30 Again the statistical robustness in the absence of the biological context is of limited value, suggest revision.

Line 39 suggest revision “is primarily associated with M. bovis”

I would suggest limiting the use of “causes” and its derivatives in the context of complex diseases. Rather I would suggest “associated with” or “strongly associated with”, as causality can be very difficult to attribute with absolute certainty for complex diseases.

Line 58 please review this sentence, it is not clear if 59% of cattle from these countries have circulating M. bovis or if 59% of countries have circulating M. bovis in their cattle.

Line 62 suggest replacing “effective” with “efficient”

Line 86 The abbreviation of “computed tomography (CT)” should be used when the term is first introduced in the abstract and the main text.

Line 179 suggest revision “overcomes the problem associated with the use of Lowenstein-Jensen solid medium.”

Note the abbreviation of this term does not seem to be required as it is only used twice.

Line 190 are there any suitable citations to verify this statement?

Line 191 suggest revision “A one health approach”

Line 198 suggest adding the country in which this study was conducted.

Line 200 suggest revision “and markets (10). While Barros”

Reviewer 3 Report

Comments and Suggestions for Authors

- Do not start with the abbreviated name of the bacteria in the title.
- Line 14: Mycobacterium bovis needs to be in italics.
- The text has several typing problems and needs to be edited. There are punctuation problems and the text is truncated in some parts, making it difficult to read. I suggest that it be reviewed by a native English speaker.
- Line 20: What is CT?
- Line 21: Mycobacterium tuberculosis appears for the first time in the abstract, so write the taxon in full.
- Line 24: M. tuberculosis should be in italics.
- Line 39: There are taxon names again without italics. I will not correct this aspect from now on. I ask that the authors at least take the time to do a detailed review of the text, correctly organizing the taxonomic nomenclature. - The article does not comply with the journal's standards, including the construction of the abstract, the organization of the text, and especially the way in which literature citations should be made.
- Lines 41-43: Data on deaths and incidence are outdated. I did not understand why the authors mentioned data from 2019 if there is more recent data, including from the WHO.
- Line 51-53: Information about the African continent is unnecessary and out of context, as it does not correlate with the methodological basis of the study, nor does it have any connection with the subject. Remove.
- In line 86, the authors describe the acronym CT, after several mentions in the text. In my opinion, the authors used parts of the text from a scientific project, without due updating and review. That is why there is data from 2019, and in several parts the text seems like a patchwork quilt. The authors should be a little more careful with the data that certainly took work to build.
- Were the treatments different? Why? The text needs explanations. - Where is the sample calculation? Is the number of patients worthy of the conclusions reached? Based on what?
- SPSS is a paid program that requires a license to use, usually included in the methods. We ask that you include the license. If there is none, I recommend that the calculations be redone in R and that the prompts used be made available as supplementary material.
- The sample number is hardly reliable for the statistical analysis performed and for the conclusions of the study. I am really curious to check the sampling design presented by the authors.

- The first paragraph of the discussion is just a repetition of the results, without any interlocution with the literature.
- The second and third paragraph of the discussion, however, present data from the literature, without any interlocussion with the results of this manuscript.
- Observing the discussion of the discussion, only two paragraphs make a "discussion" in fact. The discussion needs to be rewritten. There is no way to suggest the authors much here, as it would have to teach scientific methodology. The authors have data, know the literature but could not discuss. In my view, the discussion of the study as it is can not only be adjusted. It needs to be replanted, rewritten, restructured and presented again for review.

Comments on the Quality of English Language

 The text has several typing problems and needs to be edited. There are punctuation problems and the text is truncated in some parts, making it difficult to read. I suggest that it be reviewed by a native English speaker.

Round 2

Reviewer 3 Report

Comments and Suggestions for Authors

- The text still has problems with taxonomic nomenclature, with scientific names written in full in places where the taxon has already been mentioned previously, and the genus needs to be abbreviated.
- The design and explanations about sample size need to be described in the methods.
- With the SPSS license mentioned, the R data and the prompt in the supplementary material are not necessary. As stated in the previous correction, this would only be necessary if there was no SPSS license.
- This comment was not resolved by the authors: "Observing the discussion of the discussion, only two paragraphs make a "discussion" in fact. The discussion needs to be rewritten. There is no way to suggest the authors much here, as it would have to teach scientific methodology. The authors have data, know the literature but could not discuss. In my view, the discussion of the study as it is can not only be adjusted. It needs to be replanted, rewritten, restructured and presented again for review.". It is important to highlight that the discussion still has problems, and begins by repeating results, in addition to lacking several justifications and counterpoints to the results presented.
